# Restoration and Calibration of Tilting Hyperspectral Super-Resolution Image

**DOI:** 10.3390/s20164589

**Published:** 2020-08-15

**Authors:** Xizhen Zhang, Aiwu Zhang, Mengnan Li, Lulu Liu, Xiaoyan Kang

**Affiliations:** 1Key Laboratory of 3D Information Acquisition and Application, Ministry of Education, Capital Normal University, Beijing 100048, China; xzzhang93@163.com (X.Z.); lmn1912601616@163.com (M.L.); liululu@cnu.edu.cn (L.L.); xy.maup.kang@gmail.com (X.K.); 2Engineering Research Center of Spatial Information Technology, Ministry of Education, Capital Normal University, Beijing 100048, China; 3Center for Geographic Environment Research and Education, Capital Normal University, Beijing 100048, China

**Keywords:** tilting sampling mode, optimal reciprocal cell, modulation transfer function (MTF), calibration, spectral fidelity, the least square method

## Abstract

Tilting sampling is a novel sampling mode for achieving a higher resolution of hyperspectral imagery. However, most studies on the tilting image have only focused on a single band, which loses the features of hyperspectral imagery. This study focuses on the restoration of tilting hyperspectral imagery and the practicality of its results. First, we reduced the huge data of tilting hyperspectral imagery by the *p*-value sparse matrix band selection method (*pSMBS*). Then, we restored the reduced imagery by optimal reciprocal cell combined modulation transfer function (MTF) method. Next, we built the relationship between the restored tilting image and the original normal image. We employed the least square method to solve the calibration equation for each band. Finally, the calibrated tilting image and original normal image were both classified by the unsupervised classification method (K-means) to confirm the practicality of calibrated tilting images in remote sensing applications. The results of classification demonstrate the optimal reciprocal cell combined MTF method can effectively restore the tilting image and the calibrated tiling image can be used in remote sensing applications. The restored and calibrated tilting image has a higher resolution and better spectral fidelity.

## 1. Introduction

Tilting sampling is a novel sampling method that can improve the spatial resolution of an image by changing the imaging angle between a charge-coupled device (CCD) line array in a sensor and the sensor’s moving direction [1,2]. Compared with high-mode sampling and super-mode sampling, tilting sampling only uses a single imaging CCD line array in the sensor, which is easier to manufacture and lower cost, and it can also avoid low registration accuracy problems caused by sampling with two CCD line arrays [3,4,5]. The tilting sampling can effectively improve the image spatial resolution by controlling the imaging angles in the sampling and combining with the appropriate image restoration steps [2,5].

Since the tilting sampling method was proposed, many scholars have focused on the research about the imaging angle and the restoration methods of tiling image. It has been found the aliasing in the tilting image is different from that in the normal image, which can be used to improve the details of the tilting image [5]. Traditionally, aliasing has been generated in the frequency domain [6]. Therefore, most anti-aliasing methods were developed in the frequency domain [7]. It has been proved the reciprocal cell method, which anti-aliasing in frequency–domain, can effectively remove the aliasing in the tilting images [8]. Based on the reciprocal cell methods, the optimal reciprocal cell was proposed for anti-aliasing, and it was found that the optimal reciprocal cell would get better results in the anti-aliasing step [9]. Compared with normal sampling methods, whose imaging angle is 90°, the spatial resolution can be improved around 1.64 times by using the 45° angle as the imaging angle of tilting sampling [2]. By sampling with some special imaging angles, and—using the reciprocal cell method in the image restoration step—the effective resolution of the tilting image can be improved [3,10]. The effective resolution is highly relative to the aliasing while establishing the relationship between aliasing and imaging systems [11]. The restoration based on hybrid reciprocal cell-wavelet can remove the aliasing and the noise of the image and can achieve a better image restoration effect [12]. However, the noise and blurring in the image are also the main factors that affect the resolution and quality of images [13]. Therefore, finding a suitable restoration method for tilting image is the key step to obtain a higher resolution image. MTF is widely recognized as the performance of the details about the imaging systems [14,15]. MTF, a key characteristic of the imaging system [16,17] is also an effective method to remove the blurring and noise in images. MTF, combining with optimal reciprocal cell method which has been used in the restoration of tilting images, has achieved better restoration results because of the special anisotropy of MTF [18,19]. The optimal imaging angle of the tilting imaging system has been studied, and the imaging angle of 72° has been chosen as the optimal imaging angle of the tilting imaging system while considering aliasing, field width, sampling grid and the effective resolution of tilting images [5]. However, most studies about the restoration of tilting images were based on a single band or imaging angles. Therefore, the tensor methods have been studied, and it makes the reciprocal cell methods succeed to be applied for restore the multi-bands tilting image [20].

Scholars have done much work on selecting the special imaging angles and the restoration methods of tilting images, making great achievements in tilting sampling. The work of that has greatly promoted a huge development of tilting hyperspectral super-resolution technology. However, most restoration methods of tilting images focus on the single band, which means that the studies had not studied the tilting hyperspectral imagery, even multi-bands tilting images. Moreover, they ignored the spectral distortion of the restored tilting images and the uncertain practicability of tilting hyperspectral imageries. The spectral distortion makes the restored tilting image lose the remote sensing practical meaning. Therefore, this work uses some existing mature methods to restore the tilting hyperspectral imageries to address the problems unconsidered yet and analyses the spectral differences between restored tilting images and original normal images to calibration the restored tilting image for ensuring its practicability.

This study is organized as follows: The proposed methods are introduced in Section 2. The restoration of the tilting image which combined the method of reciprocal cell and MTF is shown in Section 3 and the calibration results of the tilting image are shown in Section 4. Discussions are given in Section 5 and following by the conclusions in Section 6.

## 2. Methods

### 2.1. Tilting Hyperspectral Super-Resolution Imaging System

Tilting hyperspectral imaging is system sampling with a single imaging CCD line array that causes an angle between the imaging CCD line array and the moving direction of the sensor—called the imaging angle. The tilting sampling diagram is shown in Figure 1.

Compared with the normal methods that samples in the same field of view, the tilting sampling image method has a smaller pixel dimension and a higher sampling density so that the tilting sampling can achieve a higher resolution of the image.

The image g that sampled by the tilting hyperspectral imaging system can be expressed as:(1)$$g=\nabla \Gamma \cdot \left(\bar{F}\left(H\right)*g_{ideal}\right)+n$$
where the ∇Γ is the sampling grid of the tilting imaging system, $\bar{F}$ means the inverse Fourier transform, H is the MTF of the imaging system, $g_{ideal}$ means the ideal image before sampling, and n is the noise from the sensor.

There are some differences between the normal imaging system and the tilting hyperspectral imaging system at the imaging angle. The imaging angle of the normal imaging system is 90°. The width of the field view for tilting image was different from the different imaging angle θ, and the field of view FOV will be:(2)$$FOV=\begin{array}{cc} l\sin\theta & \theta \in \left[0,90\right] \end{array}$$
where the l is the length of the CCD array in the imaging system. The sensor we used in this study has 840 bands, its detective wavelength from 400 nm to 1000 nm and its imaging principle is shown in Figure 2a, the reflected light of objects enters the entrance slit via the objective lens, and is received by the area camera after passing through the beam split module, the data achieved by this sensor has two axes, one for spatial axis, another for spectral axis, which means the data do not only have the spatial information, but also with the spectral information [21]. While sampling (shown in Figure 2b), it should move at a constant speed and sampling a line data for an image in each sampling frequency, at the end of the scanning, the objects hyperspectral imagery data were achieved. Moreover, its parameters are shown in Table 1.

In Table 1, S/N means the signal-noise ratio.

Second, the tilting imaging system sampled with the irregular hexagonal sampling grid in the common imaging angles [8], which makes an interpolation step before image restoration, and this step will cause some noise and aliasing that cannot be removed in the restoration step. However, if we choose some special imaging angle and control the sensor moving speed, the square sampling grid could be taken in sampling and these imaging angles could be calculated from:(3)$$\{\begin{array}{c} \begin{array}{cc} c\sin\theta =\frac{c}{N}\cos\theta & 0<\theta \leq \frac{\pi }{4} \end{array}\\ \begin{array}{cc} c\sin\theta =N\cos\theta & \frac{\pi }{4}<\theta \leq \frac{\pi }{2} \end{array} \end{array}$$
where c is the detector size and N is a positive integer. Moreover, Equation (3) can be transformed as:(4)$$\{\begin{array}{c} \begin{array}{cc} \tan\theta =\frac{1}{N} & 0<\theta \leq \frac{\pi }{4} \end{array}\\ \begin{array}{cc} \tan\theta =N & \frac{\pi }{4}<\theta \leq \frac{\pi }{2} \end{array} \end{array}$$
where N is a positive integer, the calculated value of θ are θ≈18°,27°,45°,63°,72° and we can get the square sampling grid in these angles by controlling the sensor moving speed, the speed $v_{sensor}$ is [11]:(5)$$v_{sensor}=M_{object}\cdot FR\cdot \cos\theta$$
where $M_{object}$ is the real size of the object in a pixel of the image, FR is the sampling frequency of the sensor.

The image achieved from the tilting imaging system was tilted, therefore, the tilting image should have a special image geometry correction step to rectify it as a normal image. If the achieved tilting image is g(x,y), the rectifying image is $g^{\prime }\left(x_{1},y\right)$ and $x_{1}$ can be expressed as:(6)$$x_{1}=x+\left(y\tan\left(\theta \right)\right)+1$$

The offset of the different lines in the image was different, the offset for the *i*th column $offset_{i}$ can be expressed as:(7)$${\mathrm{offset}}_{i}=\left(y-i+1\right)\tan\theta$$

Moreover, then, rectify each column of the image:(8)$$g^{\prime }\left(j,i\right)=g\left(j-offset_{i}+1,i\right)$$
where j means the line of the image.

### 2.2. Band Selection of Tilting Hyperspectral Imagery via pSMBS Method

Hyperspectral imageries have characteristics of large amounts of data and data redundancy, which means that there are massive data in the hyperspectral imagery that cannot be used efficiently [22]. There are 840 bands of the image data which we used in this research. It makes some aliasing exist in the adjacent bands. Therefore, we reduced the number of bands by selecting the band that has strong independence and great information richness as the characteristic band. Then we can process fewer data and the aliasing between the near bands that can be effectively removed. The adaptive hyperspectral band selection method based on the *p*-value was proved to be suitable for our data.

Generally, the *p*-value is considered a representation of the probability of result while the hypothesis is true, the *p*-value can be expressed as a statistic t with the degree of freedom v [23]:(9)p=1−A(t|v)
where A(t|v) can be expressed as:(10)$$A\left(t|v\right)=\frac{1}{V^{\frac{1}{2}}B\left(\frac{1}{2},\frac{v}{2}\right)}{\int_{-t}^{t}\left(1+\frac{x^{2}}{v}\right)}^{-\frac{v+1}{2}}dx$$
(11)$$t=r\sqrt{\frac{M-2}{1-r^{2}}}$$
(12)v=M−2
where M is the total number of samples and we can set 0≤A(t|v)≤1 [24] and p∈[0,1]. Beta function in Equation (10) can be expressed as:(13)$$B\left(\frac{1}{2},\frac{v}{2}\right)=\int_{0}^{1}x^{-\frac{1}{2}}{\left(1-x\right)}^{\frac{v}{2}-1}dx$$
where r is the Pearson’s linear correlation coefficient. While the average of the sample set $\left\{x_{i}\right\}$ and $\left\{y_{i}\right\}$ are $\bar{x}$ and $\bar{y}$ separately, r can be expressed as:(14)$$r=\frac{\sum_{i=1}^{N}\left(x_{i}-\bar{x}\right)(y_{i}-\bar{y})}{\sqrt{\sum_{i=1}^{N}{\left(x_{i}-\bar{x}\right)}^{2}}\sqrt{\sum_{i=1}^{N}{\left(y^{i}-\bar{y}\right)}^{2}}}$$

As usual, r∈[−1,1]. In general, the higher the absolute value of *r* is the higher correlation between samples it means.

### 2.3. Optimal Reciprocal Cell Anti-Aliasing Deconvolution Operator

Almansa successfully applied reciprocal cell theory to the image processing field [8]. Its main idea was to transform the original image data into the frequency domain by Fourier transform and to constrain the image in the frequency domain by reciprocal cell methods. Then the adjacent spectra which are overlapped and dislocated were unwrapped, and finally, the image aliasing was removed.

After being converted the tilting image into the frequency domain by Fourier operation, the spectra of the image $\hat{g}$ can be expressed as:(15)$$\hat{g}=G+G_{alias}+\hat{n}$$
where G is the spectra of the original image, $G_{alias}$ is the aliasing spectra of the image, and $\hat{n}$ is the noise spectra of the image. In the frequency domain, the parts of high-frequency and low-frequency of the image were distributed in different areas, so the aliasing in different positions of the image is different [11]. A weight function $\omega \left(G,G_{alias},\hat{n}\right)$ was proposed to represent noise and aliasing [8], and it can be expressed as:(16)$$\omega \left(G,G_{alias},\hat{n}\right)=\omega \left(1,\frac{Galias}{G},\frac{\hat{n}}{G}\right)=\omega \left(\frac{G_{alias}}{G},\frac{\hat{n}}{G}\right)=W\left(a,b\right)\&\&W\left(a,b\right)\in \left[0,1\right]$$
where W(a,b) is the weight function of a and b, a is relative aliasing and b is relative noise. They can be expressed as [8]:(17)$$a^{2}\left(\xi \right)=\frac{{\left|HF\right|}^{2}{}_{alias}(\xi )}{{\left|HF\right|}^{2}\left(\xi \right)}$$
(18)$$b^{2}(\xi )=\frac{{\left|N\right|}^{2}{}_{alias}\left(\xi \right)}{{\left|HF\right|}^{2}\left(\xi \right)}$$
where ξ is the spectra of the reciprocal cell, F is the spectra of the image obtained under the ideal condition and F is usually replaced by ${\left|\xi \right|}^{-1}$ [23]. When W(a,b)=1, it means that there are no aliasing and noise in the image, which only has a little impact on the image quality; When W(a,b)=0, which has a serious aliasing and noise with a great impact on image quality.

In this study, by setting the threshold of relative aliasing and relative noise $\theta_{alias}$ and $\theta_{noise}$—the shape of the reciprocal cell after constraint can be obtained.
(19)$$S_{R-ORC}=\left\{\begin{array}{ccc} \xi :a\left(\xi \right)<\theta_{alias} & \& & b\left(\xi \right)<\theta_{noise} \end{array}\right\}$$
where $S_{R-ORC}$ is the region of optimal reciprocal cell. Without considering the Fourier system, the values of $\theta_{alias}$ and $\theta_{noise}$ should be set as 1 according to theory. However, according to prior knowledge, those values should be $\theta_{alias}\approx 0.2$, $\theta_{noise}\approx 5$ [8], at this time, the optimal reciprocal cell anti-aliasing deconvolution operator $H_{ORC}$ can be expressed as:(20)$$H_{ORC}=\left\{\begin{array}{ccc} \xi :a\left(\xi \right)<0.2 & \& & b(\xi ) \end{array}<5\right\}$$

Therefore, the anti-aliasing deconvolution in the frequency domain can be expressed as:(21)$${\hat{g}}_{de-alias}=\hat{g}*H_{ORC}$$
where ${\hat{g}}_{de-alias}$ is the image frequency spectra after removed the aliasing. The effective resolution $r_{eff}$ of sampling grid Γ [11] was used to measure the resolution in images, which is expressed as:(22)$$r_{eff}\left(\Gamma ,D^{*},H,n,f\right)=\frac{1}{{\left(2\pi \right)}^{2}}\int_{D^{*}}\omega \left(G\left(\xi \right),G_{alias}\left(\xi \right),\hat{n}\left(\xi \right)\right)d\xi$$
where $D^{*}$ is the spectra support region. Moreover, the spatial effective resolution $R_{eff}$ of the image is defined as [11]:(23)$$R_{eff}=r_{eff}{}^{\frac{1}{2}}$$

From Equation (23), the larger the $R_{eff}$ value of the image is, the lower the image resolution is. It can be seen from Formula (22) that both aliasing and noise in the image affect the effective spatial resolution of the image. Therefore, effective restoration of anti-aliasing and the de-noise image is an important factor to improve the spatial resolution of the tilting image.

### 2.4. Modulation Transfer Function of Tilting Hyperspectral Super-Resolution Imaging

The modulation transfer function of the imaging system is not only an important index to evaluate the imaging quality of sensors [24,25], but also the main method to remove the blurring and noise in the image. As a function of spatial frequency, it contains factors such as image resolution and modulation contrast, which can indicate the capability of imaging sensor object identification objectively [14].

Assuming that the natural scene which collected by the imaging system is f(x,y) and the response function of the tilting hyperspectral imaging system is h(x,y), the final image g(x,y) obtained by the imaging system can be expressed as:(24)g(x,y)=f(x,y)∗h(x,y)

In the Formula (24), * means deconvolution operator. In general, the response function of the imaging system is h(x,y), also called PSF(x,y) [26] which is the response function of the Point Spread Function System [27] and the transfer function (TF) of the system after the response function is transferred to the frequency domain H(X,Y).

The Formula (24) is transformed to the frequency domain as follows:(25)G(X,Y)=F(X,Y)∗H(X,Y)

Therefore, the restored image in the frequency domain can be expressed as:(26)$$F\left(X,Y\right)=\frac{G(X,Y)}{H(X,Y)}$$

The transfer function of the system can be expressed as:(27)$$H\left(X,Y\right)=\left|H\left(X,Y\right)\right|\cdot e^{i\phi \left(X,Y\right)}$$
where |H(u,v)| is the amplitude and ϕ(u,v) is the phase. After normalizing, the amplitude can be expressed as:(28)$${\left|H\left(X,Y\right)\right|}_{norm}=\begin{array}{cc} \frac{\left|H\left(X,v\right)Y\right|}{\left|H\left(0,0\right)\right|} & \begin{array}{cc} X=0,1,\cdots ,N-1 & Y=0,1,\cdots ,N-1 \end{array} \end{array}$$
where *N* is a positive integer and |H(0,0)| is the value in the zero-frequency domain. If |H(0,0)|=1, the normalized system MTF can be expressed as:(29)$$\mathrm{MTF}={\left|H\left(X,Y\right)\right|}_{norm}$$

If the value of point spread system function PSF(x,y) transformed by two-dimensional Fourier, equals to 1 at the zero-frequency domain, the normalized system MTF will be:(30)MTF=|H(X,Y)|

Then the restored image can be converted to:(31)$$F\left(X,Y\right)=\frac{G\left(X,Y\right)}{\mathrm{MTF}\cdot e^{i\phi \left(X,Y\right)}}$$

During processing images, the MTF value at Nyquist frequency is usually used to evaluate the imaging quality of the optical imaging system [15]. However, the result of MTF function is a curve that cannot measure imaging quality completely and MTFA can measure imaging quality better [28].
(32)$$\mathrm{MTFA}=\int_{f_{1}}^{f_{2}}\left[\mathrm{MTF}\left(f\right)-CTF_{eye}\left(f\right)\right]df$$
where $\left(f_{1},f_{2}\right)$ is the selected spatial frequency range and $CTF_{eye}$ is the special threshold minimum contrast ratio of the human eye, its value usually is 0.05 [29]. Therefore, Formula (32) can be expressed as follows:(33)$$\mathrm{MTFA}=\int_{f_{1}}^{f_{2}}\left[\mathrm{MTF}\left(f\right)-0.05\right]df$$

As usual, the higher value of MTFA is, the higher quality of the image is.

## 3. Restoration of Tilting Hyperspectral Imagery

This section has described the validation of the restoration method based on reciprocal cell and MTF for processing tilting hyperspectral imagery. Moreover, the tilting image was obtained by strictly controlling the sampling distance and the moving speed of the sensor.

### 3.1. Band Selection of Tilting Hyperspectral Imagery

The hyperspectral data used in this work have 840 bands. There is a large amount of redundancy in the data, which means a great restoration work later. Due to the influence of the sensor, the first 10 and last 40 bands of the 840 bands cannot be used because of the awful noise. Moreover, the aliasing not only exists in the single band image, but also exists in the interband of the hyperspectral imagery because of the high correlation between bands.

Considering these problems, the *p*-value sparse matrix band selection (*pSMBS*) was used in this work [22]. This method uses a *p*-value of spectral correlation to express the degree of independence between two bands of sample data rather than the band itself. The stronger the independence is, the higher the *p*-value will be and vice versa. The *pSMBS* method was used to select the bands with stronger independence (large *p*-value) and greater richness as the feature bands. The *p*-value results calculated from the original tilting image are shown in Figure 3.

In Figure 3a, the first 10 and the last 40 bands of the hyperspectral imagery were removed because of serious noise, and then, the *p*-value of it was calculated and its results were shown in (b). To verify the restoration method of tilting hyperspectral imagery, we selected some feature bands with strong independence and higher richness, which indicate by *p*-value. From Figure 3b, we can find that higher *p*-value band ID concentrated between 250 and 450, while the *p*-value higher than 0.013 (the point above the red line) was greater than most bands, therefore, the Corresponding bands were selected as the feature bands to study and the number of the feature bands was 13. Moreover, the information of selected bands is shown in Table 2.

In Table 2, the band ID means the numbered of bands, the *p*-value was the results calculated from the original tilting image via pMSBS method, and the wavelength was for the corresponding feature bands. It is not hard to find that the wavelengths of feature bands were in the range from 587.486 nm to 683.974 nm, this phenomenon depends on the spectral reflectance properties of objects because the feature bands we selected has a greater richness which means the object has a better reflectivity in this wavelengths range. Therefore, the band ID and wavelength range will be different for different research areas.

### 3.2. Tilting Hyperspectral Imagery Restored by Optimal Reciprocal Cell Combined the MTF Method

The flow chart of optimal reciprocal cell combined MTF method is shown in Figure 4.

First, the image was restored according to the flow chart in Figure 4. The restored image is shown in Figure 5:

As seen in Figure 5, some sawteeth were found on the edge because of some small errors of the imaging angle, the shaking of the sensors while collecting image data and this work has not taken interpolation step to remove the sawteeth for avoiding the quadratic noise and aliasing which cannot be processed in the restoration step.

The details of the original titling image (Figure 5a) and the restored tilting image (Figure 5b) are shown in Figure 6.

From Figure 6, it can be easily found that the details of the restored tilting image were much clearer than that of the original tilting image visually, we can distinguish the lines in between 4 to 5 in (b), but that in (a) cannot. To directly analyze the effect of the restoration methods in this work, the evaluation index (AI) and modulation transfer function area(MTFA) were calculated and the values are shown in Table 3.

From Table 2, it can be found that the aliasing index of the restored tilting image was much smaller than that of the original tilting image in each band, which indicated that the aliasing in each band was effectively removed. Moreover, the MTFA value of each band after the restoration was higher than that of the original tilting image, which showed that the quality of each band was better after restoration. To analyze the DN-value change of the restored tilting image, the index, the ratio of prediction to deviation (RPD) [30], which was commonly used in remote sensing inversion, was introduced in this study. The value can be expressed as follows:(34)$$\mathrm{RPD}=\frac{\mathrm{SD}}{\mathrm{RMSEP}}$$
where SD means the standard deviation, RMSEP is the root mean square error of prediction and RPD was usually used to evaluate the predicted value in quantitative remote sensing restored image and the original image. The higher value of RPD represents a better result. It is commonly considered that good results can be obtained when the value is higher than 2 and an ideal result while it higher than 3, but with no prediction meaning while the value less than 1.8 [31,32,33]. First, we calculated the SD and RMSEP between the original tilting image (Figure 5a) and the restored tilting image (Figure 5b) and then get the RPD value. The parameters of each band calculated by the Formula (34) are shown in Table 4.

From Table 4, it was found that the RPD values of the original image and restored image were in the range from 2.3067 to 2.6915. All of those values were greater than 2, which showed that the DN-value change of the restored tilting images was consistent as we expected.

## 4. Calibration of Restored Tilting Hyperspectral Imagery

This section describes the calibration methods of the restored tilting image. There were two sets of field scenes which were designed and obtained by tilting sampling mode and normal sampling. One set of scenes was used to verify the spectral fidelity and solve the problem of spectral distortion, while another was used to test the feasibility of the restored tilting hyperspectral data in remote sensing applications. In this work, we strictly controlled the sampling modes in the same distance and the moving speed of the sensor.

### 4.1. Calibration of Restored Tilting Hyperspectral Imagery

The hyperspectral imageries were obtained by normal sampling and tilting sampling mode and the tilting images after the geometric correction are shown in Figure 7.

First, we selected 13 bands as feature bands from those 2 sets of data by the *pSMBS* method. The same size image of the interest regions was selected from two field scenes, respectively. The final selected regions of interest area in this work are shown in Figure 8.

From Figure 8a,c both images were the same size, but the field view of (c) was smaller, which means the resolution of the tilting image was higher than that of the normal image. It shows the image resolution was improved to a certain extent by only changing the sampling mode because the tilting sampling reduced the sampling interval and increased the sampling density. Two sets of interesting area images in Figure 8 were restored by optimal reciprocal cell combined MTF method. Finally, these restored images are shown in Figure 9.

From the point of view, the images in Figure 9 were much clearer, which fully showed the good effect of this method. However, the brightness of the restored tilting image was slightly getting higher than that of the original tilting image. Therefore, in this work, we have selected three corresponding regions both from the original normal image (Figure 8a) and the restored tilting image (Figure 9c) randomly, to calculate the mean DN-value and get the curves of the mean DN-value. The final mean value curves of the DN-value are shown in Figure 10.

It was found from Figure 10 that the mean DN-value trend of tilting image was the almost same as that of the original normal image, the mean value of restored tilting image pixels was higher than that of the original normal image. The results showed that there were a little spectra distortion in the restored tilting image.

To solve the distortion problem of tilting hyperspectral data found in the previous section, it needs to ensure that the spectra of the restored tilting image are distorted compared with that of the original normal image. Here, the calibration method for the restored tilting image was adopted. Therefore, for the interest region 1 of the restored tilting image and original normal image, 36 pairs of corresponding regions with the same size of 20 * 20 were selected as the sample points. The pixel values in the restored tilting image were taken as the observation values and that in the original normal image as the reference value and the least square method was used to solve the calibration equation of each band. The research flow chart in this work is shown in Figure 11.

According to the flow chart in Figure 11, the linear equation fitting maps of 13 bands were obtained and as shown in Figure 12.

It was found from Figure 12 that there was a strong linear relationship between the original normal image and restored tilting image in the DN mean value of each band. According to the pixel mean-value of the same terrain blocks between the original normal image and restored tilting image, the corresponding relationships were established. The linear equations (calibration equation) of those 13 bands were solved by the least square method:(35)$$y_{1}=0.7332+0.91840x_{1}$$
(36)$$y_{2}=-1.62060+0.94084x_{2}$$
(37)$$y_{3}=-0.00846+0.94878x_{3}$$
(38)$$y_{4}=0.05884+0.93780x_{4}$$
(39)$$y_{5}=1.22103+0.93155x_{5}$$
(40)$$y_{6}=0.58365+0.93949x_{6}$$
(41)$$y_{7}=0.73937+0.96379x_{7}$$
(42)$$y_{8}=-1.40800+0.95006x_{8}$$
(43)$$y_{9}=-3.41084+0.93125x_{9}$$
(44)$$y_{10}=-2.40987+0.94204x_{10}$$
(45)$$y_{11}=-0.38730+0.93837x_{11}$$
(46)$$y_{12}=0.61855+0.91413x_{12}$$
(47)$$y_{13}=-0.15475+0.90989x_{13}$$

According to the calibration equations, the restored tilting image was calibrated. The calibrated image is shown in Figure 13.

Form Figure 13, it was not hard to find that the (c) is much clearer than (a) and (b) visually. Compared with the reference value, the RPD values of the calibrated tilting image (Figure 13c) and the original normal image (Figure 13a) were calculated. There were the evaluation indices shown in Table 5:

R^2^ was the goodness of fit obtained by the least square method.

From Table 5, it can be found that the maximum value of RPD was 5.1937 and the minimum value was 2.4436. All of the values were greater than 2 and some of the value was higher than 3. Furthermore, the minimum value of R^2^ is 0.9930 and the maximum value is 0.9954, which showed that the fitting results of calibrated equations were very good. In conclusion, the spectral fidelity of the calibrated tilting image was better. The Contrast graphs of the mean DN-value between the calibrated tilting images and the original normal images are shown in Figure 14.

From Figure 14, it was found that the pixel mean-values of each band in the calibrated tilting image was not only consistent in trend with the original normal image, but was also very close in value for most points, which showed that the spectral fidelity of calibrated tilting image became better. The calibrated tilting image not only can get higher resolution, but also undistorted spectra.

### 4.2. Classification of Calibrated Tilting Hyperspectral Imagery

This section describes the application of the calibrated tilting image. For this purpose, the calibrated tilting image was used to classify and compare the results of calibrated tilting images and original normal images to prove whether tilting images can be used in actual application in remote sensing. This research chose interesting area 2, which has more distinguishable in features, as the research area. The image was restored and calibrated by the method we selected in this study. Moreover, then, the K-means classification method was applied to classify the calibrated tilting image and original normal image which is less affected by artificial factors. The classification results are shown in Figure 15.

From Figure 15, it can be found that the leaf of both (a) and (b) were distinguished from the background. Moreover, there was no significant visual difference between those two images. The evaluation accuracy of the classification results was counted and shown in Table 6 and Table 7.

It can be seen from those two tables that the overall classification accuracy of the original normal image was 96.2848%, and the Kappa coefficient was 0.9365. The overall accuracy and Kappa coefficients of the calibrated tilting image were 98.1016% and 0.9645, respectively, which were higher than that of the original normal image. The reason was that the resolution of the calibrated tilting image was higher. It is fully shown that the calibrated tilting image can be used in the actual application of remote sensing.

## 5. Discussion

In this study, we try to use the existed method the reciprocal cell combined MTF to restore the tilting hyperspectral image and find a possible way to ensuring the spectral fidelity of the restored tilting image.

Since the tilting sampling was proposed, it was aimed to achieve a higher resolution hyperspectral image [1,2,3] and many restoration methods have been developed for tilting image. The aliasing in tilting image was different with that in normal image [11], and the reciprocal cell method has an excellent effect in anti-aliasing step [3,19]. Therefore, the reciprocal cell method was developed to anti-aliasing of tilting image [9,20]. However, the reciprocal cell method has ignored the noise and blurring in the tilting image [5]. Furthermore, the MTF was introduced to de-noise and de-blurring of tilting image. Therefore, the reciprocal cell combined with the MTF method has widely used to restore the single band tilting image. Furthermore, this study chose this method to restore the tilting hyperspectral image. The restored results showed that the image quality has been improved after restoration, and the evaluation index showed that the DN value change was consistent as we expected. It means that the reciprocal cell combined MTF method can effectively restore the titling image.

The spectral fidelity of the restored tilting image is an unconsidered problem because most research studied the single band tilting image. This study focused on building the model between the restored tilting image and the original normal image and then, solved the calibrated by the least square method. The results showed that the classification accuracy of the calibrated tilting image has not declined, which means that the proposed calibrated method can ensure its result can be used in the remote sensing application.

For this study, it was successful to study on the tilting hyperspectral or multispectral tilting image and consider the problem of spectral distortion and practical availability in remote sensing application after restoration. It not only verified the traditional restoration method of tilting image, which named optimal reciprocal cell combined MTF method, can effectively restore the tilting hyperspectral imagery, but also proposed the calibrated method of restored tilting image and ensured its practicality in remote sensing application.

However, the weight of aliasing, noise and blurring in the tilting images was different, but this study has assumed those are at the same level while the restoration step. Therefore, our group will still work on the restoration method of tilting image to achieve better results of it.

## 6. Conclusions

This study used the traditional restoration method optimal reciprocal cell combined with MTF to restore the tilting hyperspectral imageries and calibrated the restored tilting image to solve the problem of spectra distortion by the calibrated equation which solved via the least square method. By comparing the classified results between calibrated tilting images and original normal images, it is not hard to find that the classification accuracy has not declined which means the calibrated tilting image can be used in the actual remote sensing applications. The results showed that the optimal reciprocal cell combined with MTF can be used to restore the tilting hyperspectral imageries and the proposed calibrated method can ensure the spectrum of the tilting image. In summary, the tilting image restored by the reciprocal cell combined MTF with the method and calibrated its results by the proposed method has a higher spatial resolution and better spectral fidelity.

## Figures and Tables

**Figure 1 sensors-20-04589-f001:**
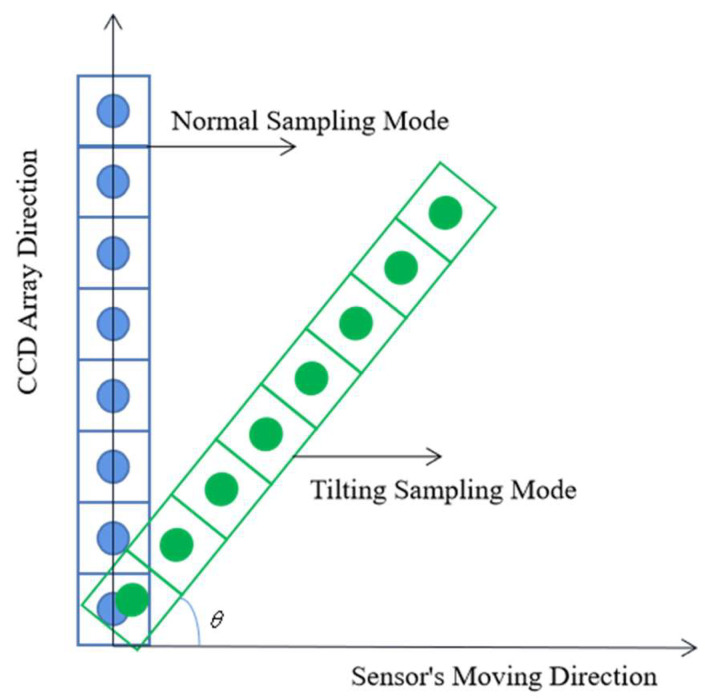
Tilting sampling and normal sampling.

**Figure 2 sensors-20-04589-f002:**
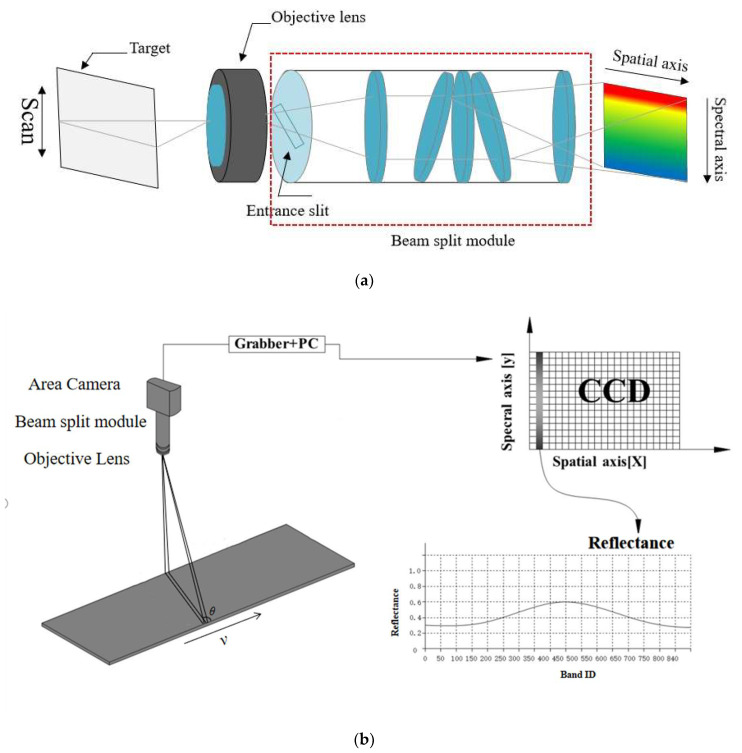
(**a**) Imaging principle of the sensor we used; (**b**) sampling diagram of the sensor we used.

**Figure 3 sensors-20-04589-f003:**
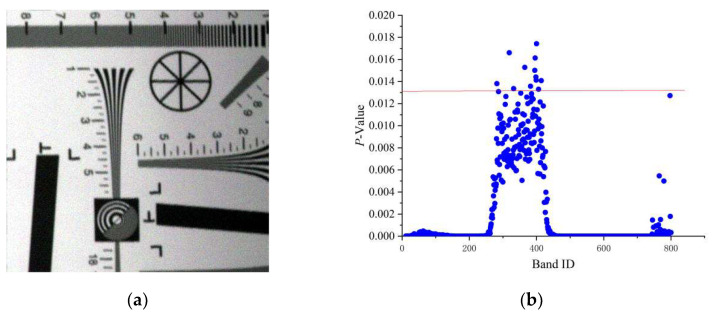
(**a**) Original tilting image; (**b**) *p*-value distribution map.

**Figure 4 sensors-20-04589-f004:**
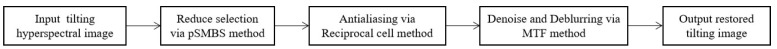
Flow chart of the optimal reciprocal cell combined the modulation transfer function (MTF) method.

**Figure 5 sensors-20-04589-f005:**
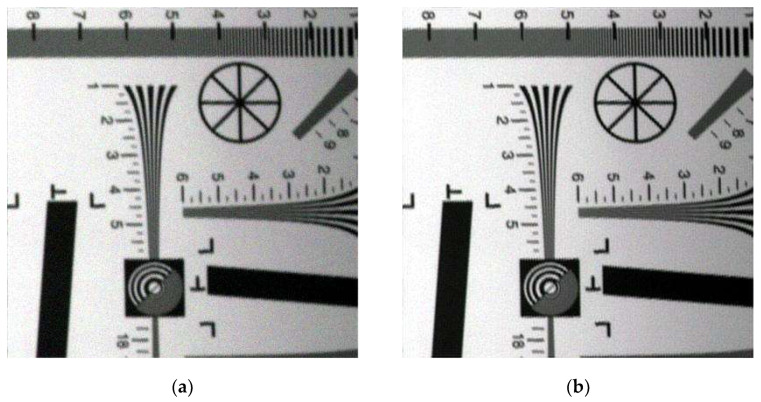
Pseudocolor tilting image. (**a**) Original tilting image; (**b**) restored tilting image.

**Figure 6 sensors-20-04589-f006:**
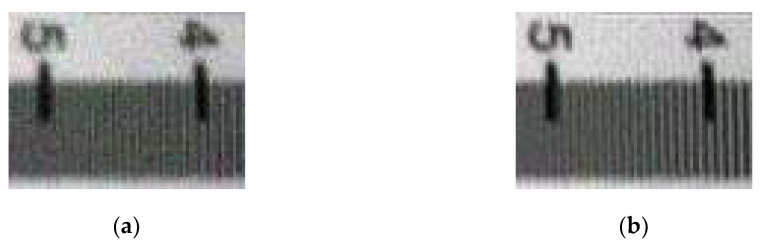
Details of the original tilting image and the restored tilting image. (**a**) Details of the original tilting image; (**b**) details of the restored tilting image.

**Figure 7 sensors-20-04589-f007:**
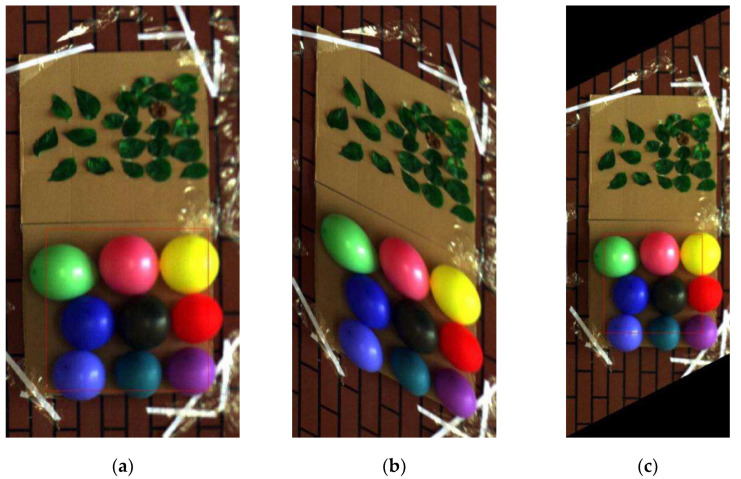
Image acquired by normal or tilting sampling. (**a**) Normal image; (**b**) tilting image; (**c**) tilting image after geometric correction.

**Figure 8 sensors-20-04589-f008:**
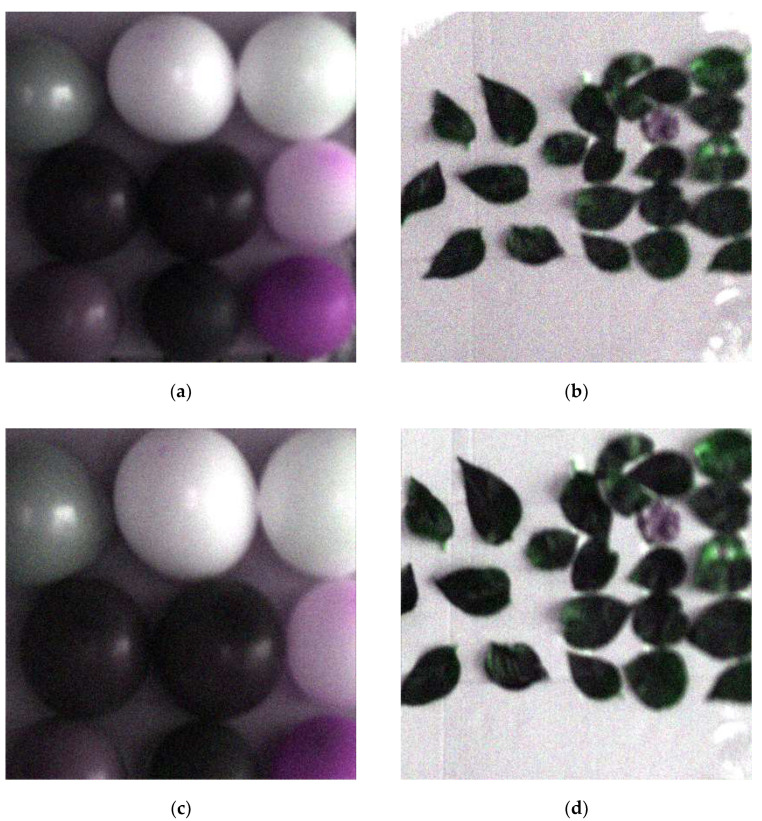
Pseudocolor image of interest region. (**a**) Normal image of interest region 1; (**b**) normal image of interest region 2; (**c**) tilting image of interest region 1; (**d**) tilting image of interest region 2.

**Figure 9 sensors-20-04589-f009:**
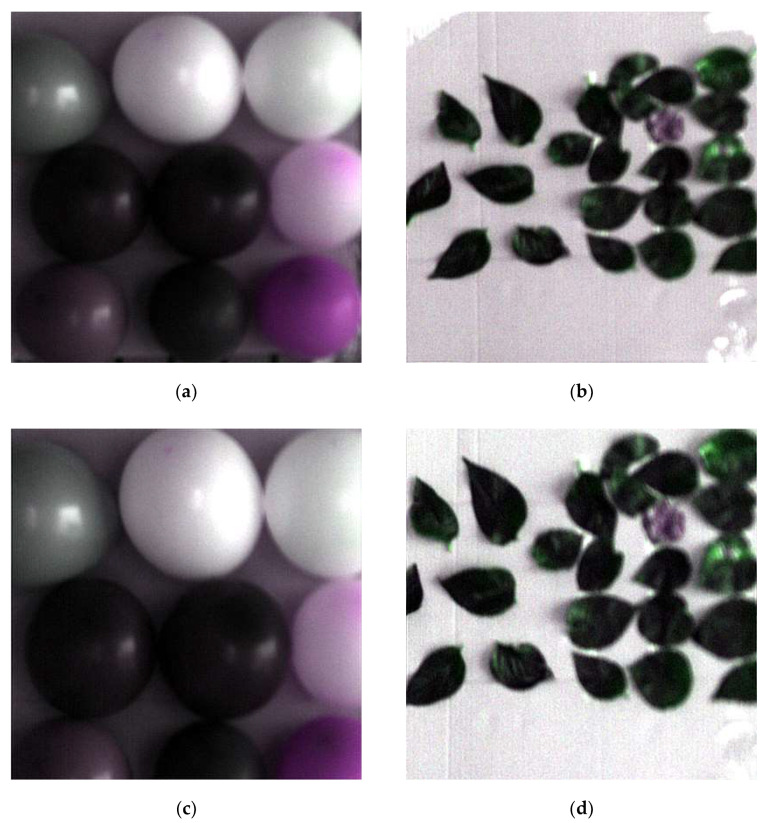
Restored pseudocolor image of the interesting area. (**a**) Restored normal image of interesting area 1; (**b**) restored normal image of interesting area 2; (**c**) restored tilting image of interesting area 1; (**d**) restored tilting image of interesting area 2.

**Figure 10 sensors-20-04589-f010:**
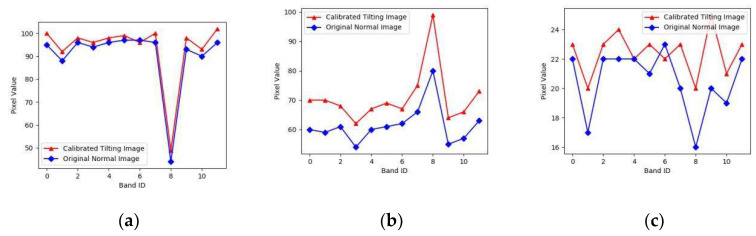
DN Mean-value curve of the same object area both of normal image and restored tilting the image. (**a**) Mean-value curve of corresponding area 1; (**b**) mean-value curve of corresponding area 2; (**c**) mean-value curve of corresponding area 3.

**Figure 11 sensors-20-04589-f011:**
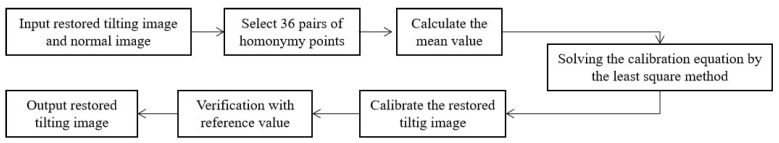
Flow chart of tilting image calibration.

**Figure 12 sensors-20-04589-f012:**
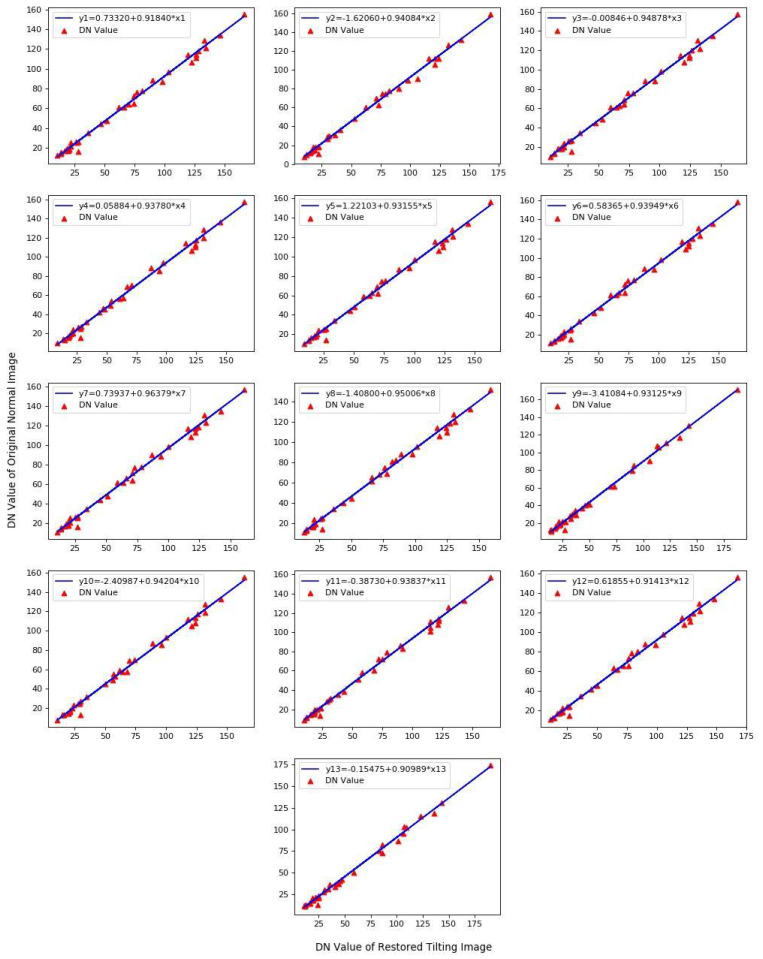
Calibration equation fitting of each restored tilting image band.

**Figure 13 sensors-20-04589-f013:**
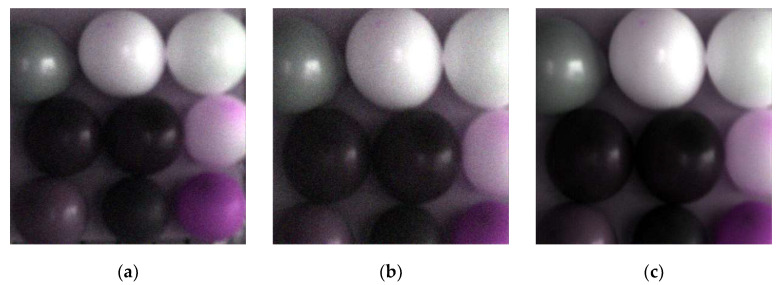
Pseudocolor image of tilting image and normal image. (**a**) Pseudocolor original normal image; (**b**) pseudocolor original tilting image; (**c**) pseudocolor calibrated tilting image.

**Figure 14 sensors-20-04589-f014:**
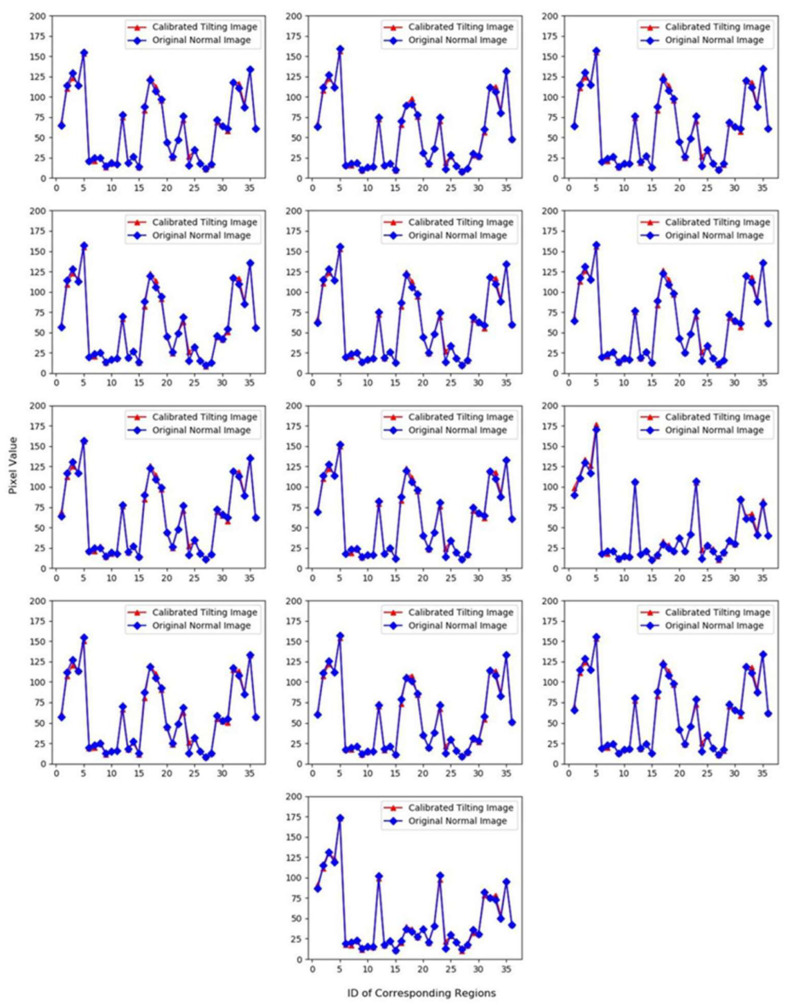
Comparisons maps of mean-value between the calibrated tilting image and the original normal image.

**Figure 15 sensors-20-04589-f015:**
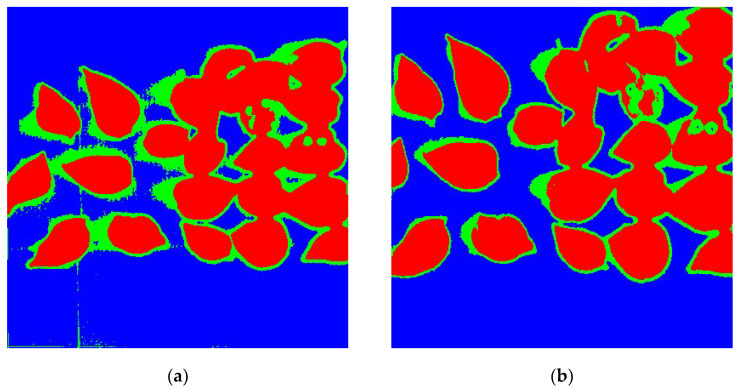
Classification results of original normal images and restored tilting images. (**a**) Results of the original normal image; (**b**) results of calibrated tilting image.

**Table 1 sensors-20-04589-t001:** Parameters of the Sensor.

Name	Parameter
Focus	23 mm
Pixel size	7.4 um
Length of CCD array	1600 (max)
Sampling frequency	33/15 fps
S/N	60 dB
F-number	2.4

**Table 2 sensors-20-04589-t002:** Information of selected bands.

ID	Band ID	*p*-Value	Band Wavelength (nm)
1	400	0.01742	673.665
2	319	0.01661	614.355
3	396	0.01613	670.722
4	365	0.01528	647.962
5	395	0.01500	669.987
6	398	0.01441	672.1934
7	399	0.01412	672.929
8	414	0.01408	683.975
9	282	0.01381	587.486
10	382	0.01357	660.433
11	332	0.01336	623.831
12	406	0.01330	678.081
13	287	0.01308	591.108

**Table 3 sensors-20-04589-t003:** Evaluation index (AI) and MTFA value of the original tilting image and the restored tilting images.

ID	Original Tilting Image		Restored Tilting Image	
	Aliasing Index	MTFA	Aliasing Index	MTFA
1	0.5899	2.1215	0.4945	2.4936
2	0.5881	2.2001	0.4939	2.4656
3	0.5882	2.1331	0.4935	2.5387
4	0.5884	2.2426	0.4928	2.5306
5	0.5888	2.1365	0.4964	2.5427
6	0.5884	2.1717	0.4947	2.5395
7	0.5890	2.1053	0.4963	2.4817
8	0.5892	2.1901	0.4935	2.5288
9	0.5888	2.0792	0.4948	2.4100
10	0.5877	2.1785	0.4951	2.5089
11	0.5889	2.1687	0.4957	2.5400
12	0.5886	2.1259	0.4931	2.4916
13	0.5877	2.0730	0.4949	2.4199

**Table 4 sensors-20-04589-t004:** Ratio of prediction to deviation (RPD) value of the original tilting image and the restored tilting image.

ID	SD	RMSEP	RPD
1	64.7079	24.9940	2.5889
2	63.0294	23.4184	2.6915
3	66.2383	24.8980	2.6604
4	65.1428	26.9604	2.4162
5	65.6546	24.9530	2.6311
6	65.1358	24.6239	2.6452
7	64.5875	25.2995	2.5529
8	64.8738	28.1241	2.3067
9	61.4836	24.2497	2.5354
10	65.3748	25.5949	2.5542
11	64.1214	25.4107	2.5234
12	64.5751	26.7004	2.4185
13	61.3591	23.9015	2.5672

**Table 5 sensors-20-04589-t005:** Evaluation Indicators for calibrated images and indicators of the goodness of fit.

Band ID	RMSE	SD	RPD	R^2^
1	10.4722	43.0816	4.1139	0.9941
2	8.8889	43.2790	4.8689	0.9948
3	11.9998	43.6206	3.6351	0.9939
4	12.3611	43.4579	3.5157	0.9930
5	12.7222	43.3476	3.4072	0.9932
6	11.7500	44.1619	3.7585	0.9938
7	17.9444	43.8490	2.4436	0.9938
8	11.4444	43.5388	3.8044	0.9938
9	7.9444	41.2611	5.1937	0.9953
10	16.0278	43.4253	2.6898	0.9930
11	5.1425	43.4253	5.1425	0.9954
12	4.0982	43.8279	4.0982	0.9943
13	4.7384	41.4607	4.7384	0.9950

**Table 6 sensors-20-04589-t006:** Classification accuracy evaluation of original normal image.

Class	Leaf (%)	Background (%)	Others (%)	Total	Overall Accuracy	Kappa Coefficient
Class 1	92.45	0.00	0.00	45.51	96.2848%	0.9365
Class 2	7.55	0.00	100.00	11.92		
Class 3	0.00	100.00	0.00	42.57		
Total	100.00	100.00	100.00	100.00		

**Table 7 sensors-20-04589-t007:** Classification accuracy evaluation of restored tilting images.

Class	Leaf (%)	Background (%)	Others (%)	Total	Overall Accuracy	Kappa Coefficient
Class 1	100.00	0.00	38.46	52.82	98.1016%	0.9645
Class 2	0.00	0.00	61.54	3.04		
Class 3	0.00	100.00	0.00	44.14		
Total	100.00	100.00	100.00	100.00		

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
