# Peer review of "Restoration and Calibration of Tilting Hyperspectral Super-Resolution Image"

_sensors, 2020, doi:10.3390/s20164589_

Round 1
Reviewer 1 Report
Reviewer comments on the manuscript ‘Restoration and Calibration of Tilting Hyperspectral Super-resolution Image’ by Xizhen ZHANG et al.
The authors are discussing the so-called ‘tilting sampling’ method where a linear CCD sensor is scanned over the image plane of an optical system at controlled speed and with angle between the sensor long axis and the motion direction <90 degrees. Specifically the authors use a series of data processing methods developed earlier. The intent of the paper is to extend previous work onto hyperspectral imagery.
I found the manuscript’s English reasonable (bar a few sentences) but found the scientific clarity lacking throughout. I did not find it easy to track the data that are actually processed in each figures.
Moreover, I am relatively unconvinced on the outcome of the work, considering the intent here is on hyperspectral. The manuscript does not convey a clear assessment of the benefit of the hyperspectral consideration here. Perhaps there is an overall lack of benchmarking, probably weakening the work.
I would not recommend the text for publication as is.
Some specific comments to illustrate the above:
- In general CCD or CMOS sensor would be physically measuring 3 color bands (may vary) and processing of the three signals would be made to generate a hyperspectral image. The authors do not specify the sensor used but I assume this would be similar to them. I wonder if the work would not be more interesting if the processing would be done from these few raw signals rather than pre-processed hyperspectrum (which would already introduce errors in color evaluation). Hyperspectra are large data and possibly should be avoided here if the information is already in the original 3 sensed bands (assuming a 3 color sensor here), this would remove the need from the statistical elimination of the hyperbands.
- There is little consistency in the writing of equation, the same symbol being used for different meaning in different or same section sometimes (just a couple of example: convolution symbol, M). Also some symbols are not defined. This makes the manuscript rather unnecessarily difficult to follow.
- Unclear what are the data/images used to build up Figure2? The same for the table on RPD etc.
- Table 2 is presented but there is little possibility of the reader to grasp the meaning without benchmark. The authors processing seems to changes the metric values somewhat but proper benchmarking I needed to assess the magnitude of the improvement claimed. Same of table with RPD values: some values are marginally above 2.5 some are not; how much above 2.5 must the values be for the improvement considered significant? Again, the difficulty may be linked to a lack of benchmarking.
- Figure 9, what are pt1,2 and 3. There is a general lack of description of what are the data in the figures throughout.
Reviewer 2 Report
This work reports the tilting sampling aiming to achieve high-resolution hyperspectral imaginery. The conclusion was supported by the research data properly. It can be accepted as it is.
Author Response
Thank you very much for your kind work and consideration on accepted our manusript.
Reviewer 3 Report
The paper shows an ineresting methord for dealising data, which proves goodbresults.
The theoretical description and the given examples seem coherent, albeit not checked in full detail.
The English is verypoor and deserves a mayor revision.This is the only for an immediate publication.
Author Response
Thank you for the constructive suggestion, and we have check some grammer error in the manuscript. Due to the time limited of this reversion, maybe the revised manuscript also have some problem in English, we will still to improve the Enlish of our manuscript by seek help from native speakers of English. Please see the revised manuscript in the attachment.

Reviewer 4 Report
The topic of the article is significant, but the manuscript requires a minor review.
Specific comments:
line 28 It has been written "These results". What results? Results of classification?
lines 36-43: missing reference
line 229: Why were 13 bands selected? Why was P-value higher than 0.013 chosen?. It is not explained in detail.
The wavelengths of selected bands were in the range of 587.486145 - 683.97467. What could be the reason for this range?
line 281, Figure 6. The way of the image geometry correction should be described.
Figure 9. What does "Band ID" mean in the x axis? There is ID 0. Should the bands be from 1 to 13? It is incomprehensible.
Section 5 Discussion does not contain any literature references. Most of the discussion looks like a summary, not an explanation of the obtained results and their discussion against the background of the available literature.
Abstract and Conclusions should be more distinguished from each other.
Round 2
Reviewer 1 Report
I thank the authors for taking the time to change their manuscript and prepare a response to my comments.
Most of my concerns are addressed in these 2 documents, put together.
However, I can see that the authors explained the experimental setup in the response letter but did not include anything on this in their manuscript. I think this is needed. Also description/manufacturer of the parts used should be provided. A statement citing a reference where this can be found would at the very least be needed, if this has been reported before by the authors. The optics could be better described so that others would be able to rebuild in the future.
In addition, some numbers are probably wrongly presented. For example the wavelengths values are written with sometimes 10 digits which seems rather high (if those are correct the method of calibration should be written). The units in table 2 are probably nm and not um. I think that authors should also verify the number of significant digits used for all the other quantities derived from their data.
The authors should carefully check their manuscript for other errors of similar nature before publication.
Author Response
Thank you for your reading our manuscript and your commets. Please see the attachment,the revised portions are marked in red in manuscript.
